# Sentinel Lymph Node Techniques in Urologic Oncology: Current Knowledge and Application

**DOI:** 10.3390/cancers15092495

**Published:** 2023-04-26

**Authors:** Bartosz Małkiewicz, Paweł Kiełb, Maximilian Kobylański, Jakub Karwacki, Adrian Poterek, Wojciech Krajewski, Romuald Zdrojowy, Tomasz Szydełko

**Affiliations:** 1University Center of Excellence in Urology, Department of Minimally Invasive and Robotic Urology, Wroclaw Medical University, 50-556 Wroclaw, Poland; pk.kielb@gmail.com (P.K.);; 2University Center of Excellence in Urology, Department of Urology, Wroclaw Medical University, 50-556 Wroclaw, Poland

**Keywords:** sentinel lymph node biopsy, sentinel lymph node dissection, urological malignancies, lymphadenectomy, metastases

## Abstract

**Simple Summary:**

Lymph node (LN) metastases are a significant concern in urological malignancies. However, current imaging techniques do not reliably detect micrometastases; thus, lymph node dissection (LND) remains the gold standard. Nevertheless, there is no established ideal LND template, leading to unnecessary invasive staging and the possibility of missing LN metastases located outside the standard template. The sentinel lymph node (SLN) concept has been proposed to address this issue, but its application in uro-oncology is still mostly experimental. Nevertheless, new techniques may improve its potential. This review aims to discuss the current and future role of the SLN procedure in managing urological malignancies.

**Abstract:**

Lymph node (LN) metastases have a significant negative impact on the prognosis of urological malignancies. Unfortunately, current imaging modalities are insufficient when it comes to detecting micrometastases; thus, surgical LN removal is commonly used. However, there is still no established ideal lymph node dissection (LND) template, leading to unnecessary invasive staging and the possibility of missing LN metastases located outside the standard template. To address this issue, the sentinel lymph node (SLN) concept has been proposed. This technique involves identifying and removing the first group of draining LNs, which can accurately stage cancer. While successful in breast cancer and melanoma, the SLN technique in urologic oncology is still considered experimental due to high false-negative rates and lack of data in prostate, bladder, and kidney cancer. Nevertheless, the development of new tracers, imaging modalities, and surgical techniques may improve the potential of the SLN procedures in urological oncology. In this review, we aim to discuss the current knowledge and future contributions of the SLN procedure in the management of urological malignancies.

## 1. Introduction

The basic assumption of sentinel lymph node biopsy (SLNB) is the concept of malignant cells spreading from the primary tumor through the lymphatic vessels to a selected group of lymph nodes (LNs), called sentinel LNs (SLNs), before they reach LNs further downstream. In the beginning, SLNB was used to investigate the routes of lymphatic spread taken by malignant tumors and study the anatomy of lymphatic drainage from primary site. With time, SLNB started to be used as a method to improve nodal staging, reduce morbidity, and improve survival in some types of cancer. Currently, it is known that macrometastases (metastases larger than 2 mm) worsen the prognosis of the disease, as do micrometastases (defined as clusters of cancer cells that are between 0.2 mm and 2 mm in diameter) in the sentinel node. Research to determine the significance of micrometastases in the sentinel node is ongoing [1,2]. Penile cancer (PeCa) is a type of cancer that is ideal for further investigating the SLNB concept because of its characteristic of lymphatic spread in which distant metastases without prior involvement of inguinal LNs occur extremally rarely. In some types of cancer, such as PeCa, melanoma, or breast cancer, the role of SLNB has been established and applied in clinical practice with good outcomes for many years. For instance, in breast cancer, SLNB, rather than complete axillary lymph node dissection (LND), is recommended in clinically node-negative breast cancer; this approach results in less morbidity, such as arm swelling and shorter hospital stay [3]. Guidelines for the treatment of melanoma, as well as breast cancer, underline that SLNB should be performed in experienced centers [4]. In other malignancies, such as prostate, bladder, or kidney cancer, the role of SLNB is still undetermined, and further investigations are needed. SLN techniques are also explored in genitourinary cancers in women, such as vulvar, cervical, or endometrial cancer [5,6,7,8,9]. However, for the purposes of this study, we decided to focus on neoplasms primarily in the range of urologists’ management strategies. The aim of this review is to summarize the available data on the utility of SLN techniques in current urologic oncology, as well as to point out future directions in this field.

## 2. Markers and Imaging Methods

With the continuous development of new imaging methods and operation techniques, the procedure of SLNB has evolved in order to improve outcomes.

In the first study on SLNB, by Cabanas et al. published in 1977, the authors did not use any marker; surgeons relied only on lymphangiograms and surgical anatomical landmarks during operation [10]. Improvements in intraoperative lymphatic drainage visualization were achieved through introducing blue dye injection into the tumor [11,12]. This method had many limitations, such as a spillage effect which could worsen the visibility of anatomical structures in the operating field, and staining of the application site, which could potentially hamper the resection of the primary tumor [13]. Nowadays, there are many different types of blue dyes in use, most commonly patent blue, methylene blue, and isosulfan blue. A meta-analysis conducted by Peek et al. on the impact of different types of blue dye on the detection rate of SLN revealed that methylene blue presented the lowest false-negative rates, but no superiority of any particular blue dye could be proven in regard to detection rate [14]. The use of blue dye is associated with the risk of allergic reactions and even anaphylaxis. A recent meta-analysis and systematic review on a group of over 60,000 SLNB procedures has shown a weighed anaphylaxis rate of 0.061%, identifying isosulfan blue as the most anaphylactogenic blue dye [15].

The next step in the development of the SLNB procedure came with two studies published in 1993 on the use of a radiotracer and gamma probe for LN detection, in which the authors showed that this method is feasible and improves detection rate [16,17]. The most commonly used radiotracer is ^99m^Technetium (^99m^Tc). The advantages of using a radioisotope include good tissue penetration, enabling the detection of deeper LNs, as well as preoperative visual information (with the use of a gamma camera) and acoustic guidance during operation (generated by a gamma probe). However, significant restrictions related to the use of these markers should be noted, such as the need to have a nuclear medicine department, a specific period of time in which the procedure should be performed (related to the radiotracer decay), and exposure to ionizing radiation of medical staff (which, while maintaining standard safety procedures, is at a low, safe level) [18]. A meta-analysis conducted on a large group of patients with breast cancer has proven the high accuracy of this method in detecting SLNs, with detection rates ranging between 85 and 100% [19]. However, the combined method using a radiotracer and blue dye simultaneously is more commonly seen in practice. The data suggest higher detection rates of SLN for the combined approach, rather than for the use of a radioisotope alone [20].

To improve intraoperative visualization of SLNs, surgeons started using indocyanine green (ICG), a fluorescent marker. ICG accumulates in tissues and, in near-infrared (NIR) imaging, allows for the visualization of LNs. In comparison with blue dye, ICG has a deeper tissue penetration (though this is still very limited compared to a radiotracer and can be significantly reduced in patients with a higher body mass index) and does not stain the injection site or surgical field during tissue preparation (ICG is invisible in white light; to visualize SLNs, an additional fluorescence camera is required) [21]. In contrast to a ^99m^Tc-nanocolloid tracer, the use of ICG dye does not require a nuclear medicine facility, and there is no risk of additional medical staff exposure to radiation. Furthermore, the development of laparoscopic cameras and software facilitates better visualizing potential SLNs during surgery with the use of different light filters and image processing in real time (e.g., Figure 1). A meta-analysis on cervical cancer patients regarding the effectiveness of the method in detecting SLN reported a higher bilateral detection rate for ICG than for ^99m^Tc with blue dye; 90.3% versus 73.5%, respectively [22]. Similar results were achieved in large studies comparing ICG to other methods in endometrial cancer and esophageal cancer, but at the same time, results were inconclusive for lung cancer and colorectal cancer [23,24,25,26].

The introduction of hybrid markers, such as ^99m^Tc-nanocolloid with ICG, reduced some limitations in the application of both techniques while maintaining their advantages (e.g., deeper penetration of radiotracer and intraoperative visualization of SLNs without staining other tissues in the operation field via ICG) [27]. A study by KleinJan et al. revealed the superiority of a hybrid tracer over the blue dye technique in various malignancies. Blue dye-based detection rates were poor compared to that of the hybrid tracer (*p* < 0.001) [28]. In a recent study by Wit et al., a hybrid approach appeared to be more effective than ^99m^Tc-nanocolloid and ICG alone, with regard to the detection of PCa. The rate of metastatic fluorescent LNs was higher in the hybrid group (7.4%) compared to the sequential group (2.6%; *p* = 0.002) [29].

With the development of imaging techniques, methods such as single-photon emission computed tomography (SPECT/CT) were implemented to improve the preoperative visualization of lymphatic drainage and potential SLNs. With the aid of SPECT/CT, surgeons obtain precise, three-dimensional information on the location of LNs and their relationship to other anatomical structures, which allows them to better plan the operation and facilitate the detection of SLNs [30,31]. Due to the rising importance of prostate-specific membrane antigen (PSMA) in PCa diagnostics, positron emission tomography (PET) has been adopted as an SLN techniques as well. A recent meta-analysis including 10 studies on LN the assessment of metastases using ^68^Ga-PSMA PET/CT revealed a per-patient pooled sensitivity of 61% and specificity of 96%. However, the authors emphasized the major heterogeneity of their findings, as the sensitivity ranged from 33.3% to 96.08% [32]. Both SPECT and PET require further prospective, large cohort studies to evaluate their efficacy with various markers and to prove their accuracy. 

Currently, more papers describing a new technique using superparamagnetic iron oxide nanoparticles (SPIONs) as a tag in SLNB are being published. This new imaging technique could potentially replace radiocolloid tracers in the future, providing a nonradioactive solution. After injection around the primary tumor before surgery, SPIONs migrate through the lymphatic drainage system and accumulate in SLNs, acting as a contrast agent. LNs can be then visualized via MRI. During operation, tissues in which SPIONs are accumulated can be detected using a handheld magnetometer [33]. A recent study reported the development of a laparoscopic differential magnetometer probe, which showed promising results in terms of intraoperative SLN localization [34]. Although it is a relatively new method, the available data indicate its high efficiency in detecting SLNs. The SentiMag Pro II Study regarding the detectability of SLNs in intermediate- and high-risk prostate cancer reported a 100% identification rate, with a high sensitivity and specificity, 100% and 97%, respectively [35]. This translates into the possibility of reducing overtreatment, such as unnecessary LND. As the authors of another study demonstrated, in the case of treating patients with a 5–20% lymph node invasion risk prior to surgery, extended LND could be omitted in favor of only sentinel LND (SLND), as SPION-guided SLND is reliable in detecting almost all LN metastases in this group of cancer patients [36]. The use of SPIONs enables the more accurate identification of nodal micrometastases [37,38].

An alternative method to detect SLN first introduced in breast cancer surgery is contrast enhanced ultrasound (CEUS) using microbubbles [39]. This technique is based on the dispersion effect of microbubbles as contrast agents with sulfur hexafluoride gas, which are injected intradermally. Among the advantages of this approach is real time visualization. Moreover, this approach only requires an ultrasound machine and contrast agent—there is no need for a nuclear medicine department. Up until this point, reports of patients experiencing allergic reactions due to this approach are scarce. However, this type of imaging method is biased; hence, the visual outcome of a USG-based technique depends on the operator’s skills. Linked to this obstacle is another disadvantage of the technique, which is the long learning curve [40]. CEUS with the use of microbubbles is a new technique which still requires further investigation. A systematic review revealed that the combined sensitivity of SLN identified by CEUS in diagnosing overall sentinel nodes pathological status is 98%, specificity is 100% [41]. Further studies are necessary to standardize the method and clarify its specificity and sensitivity. As of now, there are no published studies investigating the use of this technique in the detection of SLN in urologic malignancies.

## 3. Urological Cancers

### 3.1. Penile Cancer

PeCa is a cancer that metastasizes primarily through the lymphatic pathway, first through the inguinal LNs, and then to the pelvic regions. The detection of metastatic LNs is crucial because the presence of nodal metastases is the most important prognostic factor in the survival of patients with PeCa and determines future treatment [42].

Radical inguinal lymph node dissection (ILND) is associated with significant morbidity. In a study conducted by Gopman, a wide spectrum of complications after ILND and their prevalence were described, namely the following: minor complications such as wound infection, seroma, lymphocele, wound dehiscence, cellulitis, scrotal edema, fever, or thigh numbness occurred in 65.7% of patients, while major complications occurred in 34.3% of patients and included the most severe conditions such as wound infections treated with i.v. antibiotics, skin flap necrosis, lymphocele requiring surgical intervention, nonhealing wounds, hematoma, pulmonary embolism, and sepsis [43].

The high prevalence of major complications after ILND and a high rate of overtreated patients without palpable inguinal LNs (75–80% of patients in cN0 stage did not develop metastases) initiated the need to develop a less invasive but still reliable method of nodal staging [42,44].

The first attempt to reduce the extent of lymphadenectomy without compromising oncological outcomes was the SLNB described in 1977 by Cabanas et al. on a group of patients with PeCa [10]. The promising results of this study initiated a series of studies improving the techniques of detecting SLNs and evaluating the oncological results of patients undergoing this procedure, which confirmed the benefits of its use.

This is reflected in the European Association of Urology (EAU) guidelines, which recommend performing SLNB in the case of cN0 high-risk tumors (pT1b G2 or any pT2–4), as well as individually considering SLNB in the case of intermediate-risk tumors (pT1a G2) [45]. If the SLNB shows the presence of neoplastic changes, total ILND should be performed on the same side. However, if SLNB is negative, the patient undergoes regular physician or self-examination with an optional ultrasound inspection and fine needle aspiration cytology of the inguinal LNs every 3 months (annually after 2 years). Despite the continuous development of imaging techniques, the sensitivity of noninvasive imaging methods of the affected LNs is still insufficient, which makes surgical SLNB still irreplaceable because, in up to 25% of patients in cN0 stage, micrometastases are present in inguinal LNs [45,46].

The choice of a less invasive procedure, which is SLNB compared to the classic ILND, is associated with a significantly lower complication rate (in up to 55% of patients after ILND and in about 5.7–7% of patients after SLNB) [43,47,48].

Initial studies on the reliability of SLNB showed a high false-negative rate (FNR) of 22% per patient, which means that 22% of patients with metastases in inguinal LNs were not detected with this method [49]. New research conducted in recent years aimed to develop an optimal protocol and improve the technical aspects of SLNB while maintaining the lowest possible number of complications and FNR results. Recent data showed that, along with the increase in experience of centers performing SLNB, the improvement of imaging methods, techniques of SLNB, and more detailed histopathological assessment, the FNR improved to within a range from 4.8% to 8.7% per groin (10% per patient) [27,47]. Moreover, the use of a 1-day protocol instead of a 2-day protocol can additionally decrease FNR, but with a slightly higher complication rate [50].

Simultaneously with the development of intraoperative techniques for detecting SLNs, imaging methods were improved, allowing medical professionals to plan the course of the procedure itself through the precise visualization of lymphatic drainage and more accurate determination of the SLNs. Primarily, simple lymphoscintigrams were applied to visualize SLNs. Subsequently, SPECT/CT became the new standard because of the better results in detection and more precise anatomical visualization of SLNs (due to providing 3D anatomical information) with fewer false-positive findings than planar scintigraphy [30].

Currently, the most common method of performing SLNB is by using ^99m^Tc-nanocolloid in combination with SPECT/CT, often with additional blue dye. The first step is to administer 3–4 doses of the tracer around the penis and the primary tumor the day before or on the day of the procedure depending on the implemented 1- or 2-day protocol. Then, after 10, 20, and 120 min, lymphoscintigraphy and SPECT/CT are performed. Immediately before the procedure, the location of the marker accumulation can be verified using a portable gamma camera. Intraoperatively, SLNs are detected with the use of a gamma probe (probe generates sound when a radioactive signal is detected) and then excised. Removed tissues are submitted for histopathological examination performed via serial sectioning and immunohistochemical staining to determine SLN status [51].

Since 2012, many studies describing the beneficial effect of combining ^99m^Tc-nanocolloid with ICG in relation to the increased detection of SLNs and lower FNR have been published. These outcomes emerge from the complementary properties of both markers. The results of a study by Dell’Oglio et al. confirmed the reliability and safety of using hybrid tracers with SPECT in the largest group of patients with PeCa who underwent SLNB [27]. The use of ICG–^99m^Tc-nanocolloid resulted in a significantly better visualization rate of SLNs than blue dye (95% vs. 56% intraoperative visualization rate). However, the authors of this study emphasized that the use of hybrid markers for SLNB still requires multicenter validation and greater evaluation of the cost-effectiveness of this procedure (an example of the use of ICG during SLNB is shown in Figure 2).

The use of SPIONs for SLNB in patients with PeCa was first reported by Tabatabaei et al. in 2005 [52]. In the past few years, new studies on this topic have been published, suggesting that this technique is reliable and safe but needs further investigation [53,54]. A recent study by Azargoshasb et al. reported the implementation of a hybrid ICG–SPION approach in phantom measurements, ex vivo human skin explants, and porcine models, with the intention of future application in PeCa surgery [55].

A recently published systematic review by Fallara et al. evaluated the diagnostic accuracy of dynamic sentinel lymph node biopsy (DSLNB) in a total of 2893 cN0 patients. Patients were offered further radical ILND if DSLNB revealed micrometastases. DSLNB showed a pooled weighted sensitivity of 0.50 (95% CI: 0.24–0.59), specificity of 0.82 (95% CI: 0.78–0.87), and log diagnostic odds ratio of 1.18 (95% CI: 0.29–2.97) for detecting further positive LNs at radical LND. The authors concluded that, at this point, a positive DSLNB is poorly able to distinguish further metastatic involvement upon completion of radical LND; therefore, the diagnostic accuracy of DSLNB requires further improvement [56].

De Vries et al. recently published the results of their multicenter study on the clinicopathological predictors of finding additional inguinal LNM in PeCa patients after positive DSLNB [57]. Their results corresponded to the above conclusion. Among 407 inguinal basins with a positive DSLNB, additional LNMs at ILND were present in only 64 of them (16%). A clinical prediction model based on the number of positive nodes at positive DSLNB and largest metastasis size in mm was constructed. A statistical analysis of the model revealed that it did not show clinical benefit in predicting whether to omit further surgical intervention or carry out ILND. Therefore, the completion of ILND remains indispensable in all basins with a positive DSLNB [57].

At present, further scientific research in the field of SLNB in PeCa is being conducted. This is expressed in the EAU announcement of including the results of an ongoing systematic review and meta-analysis of minimally invasive procedures for inguinal nodal staging in penile carcinoma (DSLNB and videoendoscopic inguinal lymphadenectomy [VEIL]) in the updated 2024 EAU–American Society of Clinical Oncology (ASCO) Penile Cancer Guidelines [45].

### 3.2. Prostate Cancer

Prostate cancer (PCa), unlike PeCa, represents one of the most common malignancies. It is the second most prevalent cancer and the fifth leading cause of cancer-related death in men, and it is the most frequently diagnosed cancer in 112 countries of the world [58]. One of the most significant issues in the management of PCa patients is a process of metastasis to LNs, which negatively impacts survival, influences future therapeutic decisions, and occurs in 3–42% of patients [59,60]. Extended pelvic lymph node dissection (ePLND) conducted during radical prostatectomy (RP) remains the gold-standard nodal staging procedure due to the suboptimal accuracy of preoperative imaging methods [61,62,63,64]. Intermediate- and high-risk PCa patients have a 5–70% risk of developing nodal metastases [65]. Therefore, ePLND is recommended in these groups when the risk of nodal involvement exceeds 5% [66,67]. However, the precise extent of the PLND template is still debatable, as the wide extent of lymphadenectomy may lead to a variety of comorbidities, such as lymphoceles, obturator nerve injuries, and lymphedema [68,69,70]. The oncological outcomes of PLND remain questionable, partially because nodal metastases may occur outside of the recommended PLND template [71,72,73]. SLNB would hopefully detect additional metastatic LNs, reduce the frequency of complications, and decrease the number of PLND procedures.

At this point, the EAU guidelines point toward the insufficiency of evidence supporting SLNB [66]. This state may be due to various factors, including significant the heterogeneity and inconsistency of definitions, thresholds, types of tracers, surgical approaches, detection methods, and outcome measurements [74,75,76]. SLNB in PCa was first described by Wawroschek et al. in 1999, and then investigated by the same group, as well as Rudoni et al. [77,78,79]. Now, almost 25 years later, SLNB is still considered an experimental procedure in PCa [80].

In the SLNB technique presented by Wawroschek et al., ^99m^Tc-nanocolloid was injected into the prostate before open retropubic RP. All preoperatively and intraoperatively identified LNs (‘hot spots’) are defined as SLNs [77]. Since then, novel variants of different radiotracers have been thoroughly investigated. In 1999, Motomura et al. introduced the use of ICG for detecting SLNs in breast cancer patients intraoperatively using a laparoscopic NIR fluorescence camera [81]. In 2011, Inoue et al. were the first to describe lymphoscintigraphy with ICG in PCa patients during open surgery [82]. Van der Poel et al. described the use of a hybrid tracer—a combination of a fluorescent ICG molecule with a ^99m^Tc radiotracer. The tracers were fused and injected simultaneously into the prostate [83]. The authors pointed out that the use of a hybrid tracer, in addition to preoperative SLN mapping, facilitates the intraoperative detection of SLNs. Further studies have demonstrated the overall benefits of both pre- and intraoperative detection compared to preoperative detection alone [84]. Jeschke et al. described a different approach. They applied the sequential dual-tracing method using separate injections of a radiocolloid and free ICG, allowing SLN detection both pre- and intraoperatively [85]. Other radiotracers mainly include ^99m^Tc-labeled colloids (albumin, sulfur, or phytate) and novel particles, such as SPIONs [86,87,88]. In addition to the variety of tracers, diverse navigational systems are used during SLNB in PCa, and their role is to optimize the procedure and improve detection rates of SLNs [73,84,89,90,91,92,93,94,95].

The most common method for tracer injections is the transrectal or the transperineal route. The tracer is injected into each lobe of the prostate under ultrasonographic guidance. The volume of the injected tracer depends on its type. A study by Manny et al. evaluated tracer administration routes. They concluded that robotic-guided intraoperative ICG injection was more effective than that through the cystoscope or transrectal routes [90]. A study by de Korne et al. revealed a correlation between the localization of the tracer, the location of the tumor, and SLNs. The authors concluded that tracer injection near or into the cancerous tissue is beneficial for detection [96]. Nevertheless, the most efficient method of tracer administration remains undefined, and more studies are required to optimize the procedure.

There were three major systematic reviews concerning SLN techniques [75,97,98]. Although the authors investigated slightly different aspects of SLN techniques, they observed a low specificity (except the systematic review by Wit et al.) and overall usability but indicated promising mapping qualities.

More recent studies revealed the plausible advantages of SLN techniques incorporating novel tracers, various imaging techniques, and other procedures (namely, ePLND). A study by Claps et al. showed the staging benefit of ePLND combined with ICG-guided LND over ePLND alone. The group with additional fluorescence guidance had a higher rate of LN metastases (65.9% vs. 34.1%, *p* = 0.01) in the overall cohort and higher rates of 5-year BCR-free survival (54.1% vs. 24.9%, *p* = 0.023) in the pN+ cohort [99]. Similar results were revealed by Grivas et al. [100]. Hinsenveld et al. evaluated the additive effect of the PSMA PET/CT and SLNB combination. Blending these modalities resulted in the correct nodal staging of 50 out of 53 patients (94% diagnostic accuracy). All pN+ patients were identified [101]. A study by Fumadó et al. investigated SLNB, preceded by the identification of an index lesion (IL) via MRI, which was a target for the radiotracer injection. The sensitivity, specificity, positive predictive value (PPV), and negative predictive value (NPV) were 94.4%, 100%, 100%, and 97.8%, respectively. Negative SLNB was a predictor of negative ePLND. The false-negative (FN) rate was 5.5%, whereby one patient had nodal metastasis, despite negative SLNs. Moreover, only 10 of 18 patients (55.6%) had metastasis to SLNs only [80]. Some studies showed similar results, indicating high accuracy rates, but many of them also revealed low detection rates of all metastatic LNs, preventing the application of SLND only [102,103,104,105,106,107]. Another study evaluating the efficacy of intratumoral (IT) radiotracer injections was conducted by Wit et al. In the IT group, metastatic SLNs were 73.7% of all positive LNs compared to 37.3% in the group receiving intraprostatic (IP) injections (*p* = 0.015). The authors suggested combining IT and IP routes while performing ICG–^99m^Tc-nanocolloid injections in patients with an imaging-visible tumor [108]. Another promising direction of SLN-centered investigations is the usage of magnetometer-based techniques and SPIONs [109]. The first studies revealed very promising outcomes, although the reliability of magnetometer techniques require further, more thorough studies involving larger cohorts [35,36,110,111].

In a PCa setting, SLNB aims to improve the accuracy of positive LN detection and reduce the morbidity associated with ePLND. Additionally, this technique might allow the precise determination of individual lymphatic drainage patterns for each patient, avoiding the omission of potentially metastatic LNs outside of the traditional ePLND template.

### 3.3. Bladder Cancer

SLNB for bladder cancer (BCa) is a promising technique, but its application is limited due to the anatomical complexities of the bladder’s bilateral lymphatic drainage system and intricate mechanism of lymphatic metastasis [112]. The primary lymphatic drainage from the bladder flows toward multiple groups of LNs in the pelvis, serving as intermediate stations, and it may extend beyond the perivesical region, resulting in challenges for the SLNB technique. In 2001, the first study on the use of SLNB in patients with BCa was published by Sherif et al. Before the surgery, the authors performed planar lymphoscintigraphy and then intraoperatively used a gamma probe to detect SLNs. At least one SLN was detected in 11 out of 12 patients [113].

More recently, Zarifmahmoudi et al. conducted a meta-analysis on SLNB in muscle-invasive bladder cancer (MIBC), which included 336 patients [114]. In the analyzed studies, the authors performed ePLND after detecting and excising SLNs to verify the accuracy of the SLNB technique in nodal staging. The technical aspects of the SLNB procedure differed between studies. The main differences included the injection site (submucosal or into the detrusor muscle), used markers (radioisotope, ICG, or blue dye), and the method of detection. A crucial indicator of diagnostic performance in SLNB is the false-negative (FN) rate. Various factors can contribute to an increased FN rate, including technical difficulties during tracer injection, obstruction of lymph flow to SLNs due to the metastatic mass, tracer redirection to a non-SLN, and unilateral SLN detection despite bilateral involvement [115,116,117]. The pooled detection rate of the studies included was 90% (95% CI: 85.2–93.3%), and the combined sensitivity was 79% (95% CI: 0.69–0.89%). Overall, the authors showed that pT1–2 BCa patients with cN0 would be the most suitable cohort for SLNB [114]. Analyzing SLNB in relation to the primary tumor advancement stage, including pT3/4 stage patients, results in a decreased sensitivity of 70%. However, when this group of patients is excluded, the sensitivity of the technique increases to 93%. Therefore, SLNB mapping exhibits the highest accuracy in patients with MIBC at lower local advancement stages. This finding is in accordance with similar studies in other malignancies [118,119]. More recent studies have assessed SLN techniques in various clinical settings and concluded that they are feasible and safe in BCa, although their efficacy remains questionable [120,121,122]. In 2022, Rietbergen et al. initiated a prospective pilot study evaluating the usage of a hybrid ICG–^99m^Tc-nanocolloid radiotracer. In three patients, SLNs were outside the ePLND template, and in 53% of the patients, the preoperative SLN procedure was successful [123].

Despite revealing a high sensitivity for BCa, studies on SLNB require larger prospective trials to confirm the reliability and accuracy of this nodal staging tool and to standardize the procedure.

### 3.4. Renal Cancer

The role of LND in renal cell carcinoma (RCCa) remains controversial. Research conducted in order to establish a potential survival benefit after LND failed to do so [124]. The cause of this may be the hematogenous route of spread in RCCa. Moreover, unpredictable lymphatic drainage may hamper the detection of suspicious LNs and, as a consequence, the completeness of LND and its outcome. The majority of patients in the pN+ stage simultaneously acquire hematogenous metastases [125]. Thus, the presence of metastatic LNs may indicate a systematic spread of disease, which significantly worsens survival prognosis in patients with RCCa.

SLNB can potentially improve the detection of metastatic LNs. The first two studies by Bex et al. on SLNB in patients with cT1–2N0M0 RCCa were conducted with the use of ^99m^Tc-nanocolloid injected percutaneously under ultrasonography guidance into the primary tumor the day before surgery [126,127]. After the injection, lymphoscintigraphy and SPECT/CT were performed. During surgery with the use of a gamma probe and gamma camera, the surgeons localized and then removed SLNs with adherent lymphatic tissues. In the case of the first study, all nodes could be visualized with the use of SPECT/CT, whereas in the case of the second study, SPECT/CT failed to detect nodes in approximately 30% of cases. Nevertheless, it outperformed planar lymphoscintigraphy in lymphatic mapping. The authors concluded that SLNB in RCCa is feasible and safe, bringing new data on lymphatic drainage in this type of tumor, which could be useful for planning the extent of LND. Nevertheless, the non-visualization of lymph nodes, up to this day, remains one of the limiting factors for broad the clinical implementation of the SLNB procedure using SPECT/CT in RCCa, as shown by another study investigating the possible factors influencing detection rate [128].

The issue of isolated metastatic LNs in the mediastinum, frequently observed in RCCa, was the cause of further investigations on the lymphatic spread in this type of cancer. A study by Brouwer et al., in which the authors examined lymphatic drainage from RCCa using intratumorally injected ^99m^Tc-nanocolloid followed by lymphoscintigraphy and SPECT/CT imaging, showed that, in about 18% of patients with visualized SLNs, a direct drainage throughout the thoracic duct occurred, omitting LNs [129]. In the largest prospective study investigating the pattern and location of lymphatic drainage from renal tumors using a radiotracer by Kuusk et al., the authors detected supra-diaphragmatic SLNs in 20% of patients [130]. These results may support the hypothesis of an important role of lymphovenous connections between renal tumors and the thoracic duct in the unpredictable systemic spread of RCCa.

ICG fluorescent lymphography is a technique which potentially could improve SLN detection in RCCa. The promising results of the application of ICG in other types of cancer make it worth investigating in RCCa patients [131].

Hybrid tracers such as ICG combined with ^99m^Tc-nanocolloid and used with SPECT/CT before surgery present promising results in patients treated for PeCa. To date, there is no study investigating this very promising approach in patients with RCCa; thus, its impact here remains to be seen.

SLNB with locoregional LND in RCCa seems to be a surgically safe procedure, with no significant variations in complication rate across different surgical techniques [132]. However, due to the lack of large prospective studies, SLNB in patients with RCCa currently has no clinical implication. Conclusions from studies in which patterns of lymphatic drainage from renal tumors were investigated to apply the SLNB procedure cannot yet be implemented in clinical practice, and perhaps never will be, because of the very complex way in which RCCa metastasizes.

### 3.5. Testicular Cancer

SLNB in patients with testicular germ cell tumor (TGCT) was first described in 2002. The authors of two different studies on clinical stage I TGCT patients demonstrated the feasibility of this method [133,134]. After the injection of a single dose of ^99m^Tc-nanocolloid into the funiculus or testicular parenchyma around the tumor, lymphoscintigraphy was performed. During laparoscopic SLNB, an endoscopic gamma probe was used to facilitate the detection of SLNs. In a study by Tanis et al., in two patients, patent blue dye was additionally injected intratesticularly during surgery, with no discoloration of the detected SLNs. After the initial funicular administration of the radioactive tracer, lymphoscintigraphy showed lymphatic drainage only to the inguinal region, meaning that the injection site had to be changed and the injection had to be administered intratesticularly [133]. Intratesticular injection enabled the visualization of SLNs in the para-aortic region (especially in the cases of left testis tumors), as well as the inter-aortocaval, paracaval, or common iliac region (when primary tumors were in the right testis) [134].

In a study by Satoh et al., SLNB was investigated in a group of 22 patients with clinical stage I TGTC and was the first in which all participants underwent laparoscopic retroperitoneal PLND (RPLND). The radiotracer administration protocol was similar to previous studies except for the time of tracer administration, which in this case, was always 24 h before surgery. SLNs were detected in 95% of patients, and nearly all were localized on the ventral or lateral side of the vena cava or at the aorta between the levels of the aortic bifurcation and under the levels of the gonadal arteries. Two micrometastases were found out of a group of 22 patients: 1 case of seminoma and 1 case of nonseminoma. These two patients were treated with two cycles of BEP without relapse for 29 months (seminoma) and 31 months (non-seminoma) of follow-up. All dissected non-sentinel retroperitoneal LNs were free of metastases [135].

Further studies using SPECT/CT and the portable gamma camera revealed that the use of an intraoperative gamma probe can improve the detection of additional SLNs by up to 20% [136].

Currently, in the largest study conducted by Blok et al., occult micrometastases were detected in 13% of patients diagnosed with TGCT in clinical stage I who underwent SLNB. The early detection of nodal metastases allowed for the early initiation of adjuvant systemic treatment. All patients with TGCT were without evidence of disease at a median follow-up of 63.9 months [137].

The excellent treatment results of TGCT, with 5- and 10-year overall cancer-specific survival rates close to 100%, indicate that there is no urgent need to improve the methods of treatment and diagnostics in these patients [138]. Moreover, the currently accepted scheme of surveillance after surgical treatment allows for the early detection of disease relapse or lymphatic spread. Additionally, in the case of disease recurrence, due to the high chemosensitivity of this type of tumor, it can be treated with good results using systemic therapy, but this adjuvant therapy also has potential serious short- and long-term side-effects. The use of SLNB could allow for a better selection of patients being eligible for adjuvant treatment; in patients with detected occult nodal metastases, it would allow for the commencement of treatment at an early stage, while in patients with negative biopsy, it would enable the safe activation of active surveillance. This approach could result in an overall reduction in complications associated with adjuvant therapy by lowering the risk of relapse through early treatment and of unnecessary adjuvant therapy in patients with negative biopsy. Another potential benefit of applying SLNB may be the reduction in the frequency of CT scans in biopsy negative patients, which given the young age of patients, may be beneficial due to the significant reduction in radiation exposure. Due to the limited amount of research on this issue and potential complications resulting from RPLND, further work is necessary to be able to consider changes in the currently adopted procedure and recommendations.

Continuous investigation regarding the use of the SLN technique in TGCT is in progress. A recently published paper by Zarifmahmoudi et al. described a series of nine patients with a history of non-seminomatous testicular cancer who underwent radical orchidectomy and systemic chemotherapy in the past but were qualified for RPLND because of the presence of residual retroperitoneal masses of size ≥ 1 cm. SLN mapping using ^99m^Tc-labeled phytate enabled the identification of SLNs with a detection rate of 66% and a false-negative rate of 0% [139]. The authors emphasized the need for further studies to improve the detection rate of SLNs in these patient groups.

Table 1 constitutes an overview of SLN techniques indications provided by the current guidelines.

## 4. Conclusions

The continuous development of imaging methods, surgical techniques, and new markers may change the role of SLN procedures in urologic oncology in the future. SLNB is a promising approach to nodal staging with the potential to improve diagnostic accuracy, reduce morbidity, and individualize treatment in patients with urological malignancies. Nevertheless, further research is needed to fully understand its potential benefits and optimize its use in clinical practice, especially considering the small study cohorts in previous investigations.

## Figures and Tables

**Figure 1 cancers-15-02495-f001:**
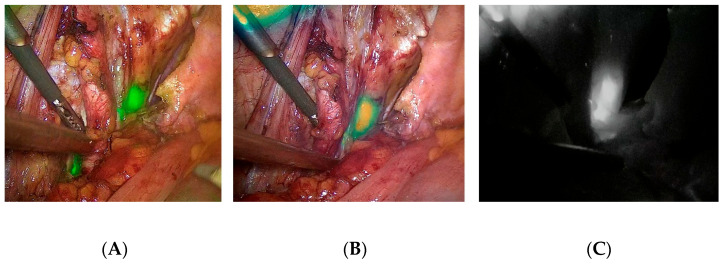
Different presentation of potential SLNs during laparoscopic RP and SLNB. LNs are marked as green or white regions: (**A**) NIR/ICG + white light overlay image; (**B**) intensity map presentation; (**C**) monochromatic presentation. Images were taken using the IMAGE1 S™ RUBINA-KARL STORZ system.

**Figure 2 cancers-15-02495-f002:**
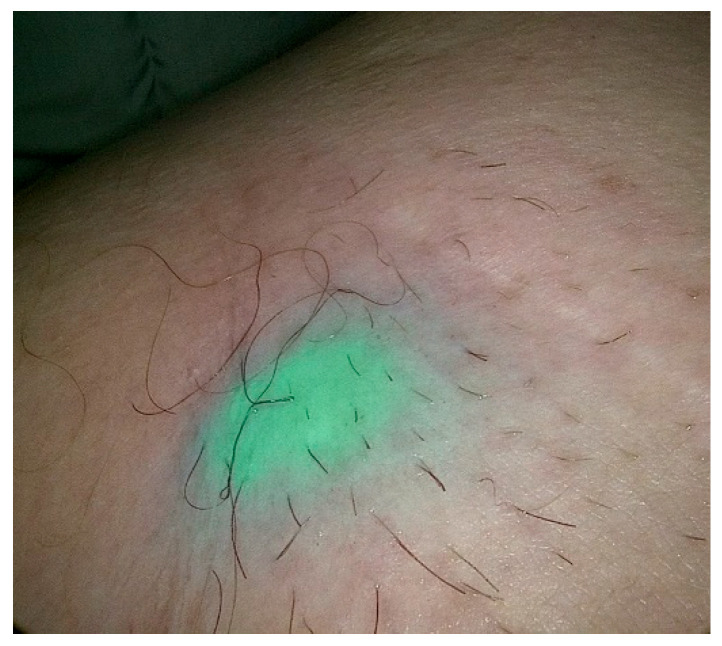
Percutaneous view of ICG marked SLN during SLNB procedure in patients with PeCa.

**Table 1 cancers-15-02495-t001:** Overview of indications concerning SLNB and SLND in urological malignancies, provided by the EAU and the AUA guidelines.

**Malignancy**	**Guidelines**	**Indications**	**References**
PeCa	EAU–ASCO	In centers that offer DSLNB as a surgical staging option, inguinal ultrasonography is obtained prior to DSLNB. In T1b or higher PCa patients, DSLNB should be offered. If DSLNB is not available and referral is not feasible, or if it is preferred by the patient after being well informed, ILND should be offered.	[45]
PCa	EAU–ASCO	SLNB remains an investigational staging strategy and lacks sufficient evidence supporting it.	[66]
	AUA	SLN techniques are not mentioned.	[140,141,142,143,144,145]
BCa	EAU–ASCO	SLNB is described among future perspectives in section concerning imaging in MIBCa.	[146,147]
	AUA	SLN techniques are not mentioned.	[148,149]
RCCa	EAU–ASCO	SLND remains an investigational technique.	[150]
	AUA	SLN techniques are not mentioned.	[151,152]
TGCT	EAU–ASCO	SLN techniques are not mentioned.	[153]
	AUA	SLN techniques are not mentioned.	[154]

EAU: the European Association of Urology; ASCO: the American Society of Clinical Oncology; AUA: the American Urological Association; PeCa: penile cancer; PCa: prostate cancer; BCa: bladder cancer; RCCa: renal cell cancer; TGCT: testicular germ cell tumor; DSLNB: dynamic sentinel node biopsy; ILND: inguinal lymph node dissection; SLN: sentinel lymph node; SLNB: sentinel lymph node biopsy; MIBCa: muscle-invasive bladder cancer; SLND: sentinel lymph node dissection.

## Data Availability

Not applicable.

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
