# Peer review of "Sentinel Lymph Node Techniques in Urologic Oncology: Current Knowledge and Application"

_cancers, 2023, doi:10.3390/cancers15092495_

Round 1
Reviewer 1 Report
This manuscript comprehends a critical literature review on the sentinel lymph node techniques in patients with urological cancers. The manuscript is interesting, with high quality figures and brings an interesting data to the urological cancer knowledge. However, authors should adjust manuscript according to the journals guidelines prior publication. Please, see few comments below:
1. Authors aimed to discuss the lymph node techniques, making the impression to be presented different techniques. However, in subheading 2 “Markers and Imaging Methods” is poorly explored. It is a superficial topic and do not explain the different markers that could ne used. It is only a little retrospective of the previous literature.
2. Authors mentioned the male reproductive cancers (i.e testicular and penile), but no female ones (such as vulvar and vagina). Maybe because little is known about these cancers. But a statement could be made.
3. Please, pay attention to the journals guidelines and adjust through the manuscript. i.e. line 203, Vries reference is lacking the number.
I have no specific comments regarding the English. Seems fine to me.
Reviewer 2 Report
Manuscript entitled "Sentinel Lymph Node Techniques in Urologic Oncology - Current Knowledge and Application"
Major issues:
1. The authors should make a comprehensive review for the deteciton technology of sentinel lymph nodes and discussing the recent advanced.
2. The size of metastatic deposits, and the significance of micrometastasis should be discussed in GU cancer.
acceptable
Round 2
Reviewer 2 Report
Manuscript entitled "Sentinel Lymph Node Techniques in Urologic Oncology - Current Knowledge and Application"
This work is acceptable for publication in the current form.
acceptable